# Sintering Copper Nanoparticles with Photonic Additive for Printed Conductive Patterns by Intense Pulsed Light

**DOI:** 10.3390/nano9081071

**Published:** 2019-07-25

**Authors:** Wan-Yu Chung, Yi-Chin Lai, Tetsu Yonezawa, Ying-Chih Liao

**Affiliations:** 1Department of Chemical Engineering, National Taiwan University, Taipei 10617, Taiwan; 2Division of Materials Science and Engineering, Hokkaido University, Sapporo 060-0808, Japan

**Keywords:** copper nanoparticle, porosity, photonic sintering, light absorption, conductive patterns

## Abstract

In this study, an ink formulation was developed to prepare conductive copper thin films with compact structure by using intense pulsed light (IPL) sintering. To improve inter-particle connections in the sintering process, a cuprous oxide shell was synthesized over copper nanoparticles (CuNP). This cuprous oxide shell can be reduced by IPL with the presence of a reductant and fused to form connection between large copper particles. However, the thermal yield stress after strong IPL sintering resulted in cracks of conductive copper film. Thus, a multiple pulse sintering with an off time of 2 s was needed to reach a low resistivity of 10^−5^ Ω·cm. To increase the light absorption efficiency and to further decrease voids between CuNPs in the copper film, cupric oxide nanoparticles (CuONP) of 50 nm, were also added into ink. The results showed that these CuONPs can be reduced to copper with a single pulse IPL and fused with the surrounding CuNPs. With an optimal CuNP/CuONP weight ratio of 1/80, the copper film showed a lowest resistivity of 7 × 10^−5^ Ω·cm, ~25% conductivity of bulk copper, with a single sintering energy at 3.08 J/cm^2^. The ink can be printed on flexible substrates as conductive tracks and the resistance remained nearly the same after 10,000 bending cycles.

## 1. Introduction

Recent advancements in printing technology have shown the capability of fabricating highly conductive patterns for light-weight flexible electronic with conductive inks. [1] Among those inks, metal nanoparticles, such as gold [2,3] and silver, [4,5] are widely used as conductive filler due to their great conductivity. To reduce cost, copper has recently attracted much attention for conductive printed films because of its low price, high conductivity and high electromigration resistance [6,7,8]. However, the high oxidation tendency of copper nanoparticles causes chemical stability problems in ink formulation and storage. Moreover, the relatively high sintering temperatures also cast difficulties in adopting copper nanoparticles for flexible plastic substrates, which usually have low melting temperatures. Therefore, preparation of copper conductive thin films from nanoparticles is still a challenging issue.

To fabricate conductive thin film patterns, there are several ink-based approaches, such as inkjet printing [9], transfer printing and e-beam lithography [10,11]. Among these processes, the inkjet printing process provides great flexibility for pattern formation and has been widely used in conductive pattern fabrication. However, inkjet printing processes require stable nanoparticles inks with high conductivity after long term and high temperature post-treatment. To resolve the high temperature sintering problem, intense pulsed light (IPL) technology has recently drawn lots of attention for metal nanoparticle sintering process. [4,12,13] IPL utilizes a xenon lamp to illuminate strong white light in a few milliseconds over a wide area. The emitted photonic energy is absorbed selectively by the printed patterns, and sinters the nanoparticles to form conductive tracks. Because of the rapid sintering process, IPL usually has low impact on plastic substrates. Generally, copper nanoparticles are protected with a polymer coating [8,14,15] to avoid oxidation, but this layer also results in large contact resistance between particles. With the nature of intense local heating, the IPL process is found to be very efficient in decomposing protective polymers, especially coupled with UV or infrared lights, [16] and IPL-sintered tracks can exhibit great conductivity close to bulk copper. Moreover, it has been found that inkjet printed copper oxide nanoparticles can absorb IPL to yield locally high temperatures and react with the reduction agent into conductive copper thin films. [17,18,19] On the other hand, the intense local heating during the IPL process can cause film damages due to large thermal yield stresses. Thus, IPL with multiple pulses [20] is found to generate more uniform temperature distribution on nanoparticle thin films and results in more effective sintering without film damages. Besides nanoparticle inks, metal organic decomposition (MOD) inks [21,22,23,24,25] have also been proposed to produce conductive tracks. These inks containing metal salts can be easily converted into metal tracks via thermal or plasma post treatments. It has also been shown that IPL post treatment can be applied on these MOD inks to fabricate highly conductive tracks [12,16,26]. However, to facilitate the chemical reactions, these MOD/IPL procedures generally need more energy consumption than copper nanoparticle inks.

The grain boundary between particles are also of critical importance to copper sintering process. After absorbing energy, copper nanoparticles need to fuse together for better electron conduction. In the literature, it has been found that cupric oxide coatings around copper nanoparticles can help particle connections in sintering process [14,27,28]. Moreover, a mixture of copper particles with various sizes are also found to be more efficient for particle fusion in the sintering process [29]. However, these experiments are performed in a nitrogen environment with a thermal sintering process. Thus, although a similar sintering approach can be adopted, particle compositions or sintering conditions for IPL process might need more adjustments to efficiently sinter the copper thin films to achieve great conductivity.

In this work, a simple ink formulation is developed to improve the light absorption efficiency in IPL for conductive copper film fabrication. First, copper particles are capped by caproic acid and preheated before used to generate a shell structure of Cu_2_O on the particle surface. With the Cu_2_O shell, necks between copper particles can be generated easily in the sintering process. In addition, a small amount of CuO nanoparticles are added in the ink to enhance light absorption efficiency. The microstructures and morphology of sintered copper films are examined carefully to understand the sintering mechanisms and to optimize the sintering parameters for best conductivity. Finally, the inks are printed on flexible substrate and sintered into conductive patterns to show potential applications of this ink for printed flexible electronics.

## 2. Materials and Methods

### 2.1. Synthesis of Copper Nanoparticles

First, 32 g of cupric oxide microparticles (Copper(II) oxide, powder, <10 μm, 98%, Sigma-Aldrich) and 3 mL of hexanoic acid (>99%, Sigma-Aldrich, St. Louis, MO, USA) were added sequentially into 400 mL of 2-propanol (>99.5%, J.T. Baker) in a 500 mL round bottom bottle under magnetic stirring at 600 rpm for 60 min. The mixture was heated to 70 °C. Then, 40 mL of hydrazine monohydrate (100%, hydrazine 64%, Acros organics, Houston, TX, USA) was added dropwise into the mixture for metal ion reduction. The reaction was terminated by putting the bottle in an ice water bath after one hour. After synthesis, the nanoparticles were collected by centrifugation at 12,800× *g*. The collected sediment was washed with ethanol (anhydrous, 99.5%, ECHO, Miaoli, TW) and centrifuged again for 5 times. The final sediment was dried under a nitrogen atmosphere.

### 2.2. Ink and Conductive Patterns Preparation

Followingly, 0.032 g of poly vinyl pyrrolidone (PVP, MW 29K, Sigma Aldrich, St. Louis, MO, USA) was dissolved into 1.2 g ethylene glycol (EG, 99.8%, Sigma-Aldrich, St. Louis, MO, USA). Then 0.8 g of the synthesized copper nanoparticles and/or 0.01 g of copper oxide nanoparticles (50 nm, Sigma-Aldrich, St. Louis, MO, USA) were dispersed in the PVP/EG solution with magnetic stirring for 1 h, and sonicated in a sonication bath (DELTA, DC300H) for 60 min. Poly(ethylene terephthalate) films (PET, Universal film, Tokyo, Japan) were cleaned by sonication in ethanol for 1 h. The substrate surface was further cleaned by oxygen vacuum plasma (Eastern Sharp Ltd., Carson, NV, USA) right before printing. The ink was printed on substrates by a robotic dispensing system (Dispersion Tech. DT-200F, TPE, Taiwan). All the samples were printed with the same printing parameters to have the same deposition rate and thus the same thickness was expected. The printed patterns were dried at 60 °C on a hot plate for 30 min to remove remaining solvent. The dried samples were then sintered by intense pulsed light (IPL, Xenon, X1100, Wilmington, NC, USA).

### 2.3. Characterization

The microstructural characteristics of copper nanoparticles were examined by transmission electron microscope (TEM, JEOL, JEM-2100F, Tokyo, JAPAN), X-ray diffraction (XRD, Rigaku, Ultima IV, Spring, TX, USA), Differential Thermal Analysis Thermoanalyzer(TG-DTA, Rigaku Thermo plus2 system TG8120, Spring, TX, USA) and ultraviolet–visible spectroscopy (JASCO, V670, Tokyo, Japan). The morphology and cross section of sintered films were analyzed by scanning electron microscope (SEM, FEI, Nova NanoSEM 230, Columbus, OH, USA). The sheet resistance (R) of copper films was measured by a four-point probe (Keithlink, TG-2, Taipei, Taiwan). Copper thin film samples (~1 cm^2^) were placed on a flat surface and gently touched with the probe. At least 5 sample points were sampled to collect the sheet resistance data.

## 3. Results and Discussion

### 3.1. Characterization of Core-Shell Copper Nanoparticles

The synthesized copper nanoparticles (CuNPs) show a core shell structure. Figure 1a shows TEM image of the CuNPs. The core particle has a diameter of 120 nm, and is surrounded by a shell of oxide. Because the capping agent used in the synthetic route has a short carbon chain, oxygen penetrates into the surface layer and reacts with the copper to generate small cuprous oxide particles of 10 nm size on the surface [27]. From the XRD pattern (Figure 1b), the surrounding oxide layer is mainly composed of Cu_2_O, which can be identified by the broad peak at 36.4° in 2θ in the XRD pattern. The TG-DTA data (Figure 1c) indicate that caproic acid totally decomposed at 160 °C, and the synthesized CuNPs start being oxidized. From the weight gain after total oxidation at high temperatures, one can also calculate that the core-shell CuNPs are composed of 97% of Cu and 3% of Cu_2_O.

### 3.2. IPL Sintering for CuNPs

The core-shell CuNPs can be quickly sintered after IPL irradiation. The UV-vis spectrum of CuNPs (Figure 2a) shows two main absorption peaks at 360 nm and 800 nm, which correspond to the Cu core particles and surrounding Cu_2_O particles [30]. These two peaks are broad due to the wide size distribution of CuNPs. Thus, these particles can absorb the IPL emission light, and the plasmon resonance after light absorption leads to heat generation. As a result, the organic PVP layers covering on the CuNPs are decomposed and the shell Cu_2_O is reduced to Cu, as the mechanism proposed by Ryu et al. [15] Figure 2b compares the XRD patterns before and after IPL sintering. The broad peak at 36° representing Cu_2_O diminishes after IPL irradiation, indicating successful reduction of oxide.

### 3.3. Effects of IPL Energy Intensity on Film Conductivity

The copper thin films become highly conductive after IPL sintering. In order to understand the CuNPs sintering mechanism and minimize the sintering energy, the copper film samples are kept at a fixed 3-cm distance from the light source and sintered with various light intensity from 2 to 3.4 J/cm^2^ with a single pulse. At low IPL energy of 2.37 J/cm^2^, the CuNPs are not totally sintered and thus a large measurement incertitude is observed (Figure 3a). As IPL energy density increases, CuNPs are well sintered and with uniform conductivity and shows the lowest resistivity of ~10^−4^ Ω·cm, or 1% conductivity of bulk copper, at 2.77 J/cm^2^. The color of the sintered copper turns from dark red to brown color with a metallic luster (Figure 3b). The increase in conductivity and changes in color indicate the low temperature sintering process of Cu/Cu_2_O particles. As proposed previously by Yonezawa et al., [28] the surrounding small Cu_2_O can form necks between CuNPs if the temperature reaches the glass transition temperature of surrounding polymers. The temperature increased in copper film with one single IPL can be roughly estimated by the following simple energy balance:(1)$$∆T=\frac{\mathrm{absorbed}\;\mathrm{energy}\;\mathrm{density}}{{\rho C}_{p}t}$$
where ρ is density, C_p_ is the heat capacity and t is the thickness of the copper thin film. Assuming that CuNPs can only absorb ~15% of the emitted light, according to the UV spectrum, the temperature can increase to ~250 °C with light intensity of 2.77 J/cm^2^, which is much higher than the glass transition temperature of PVP (around 150 ~ 180 °C). Therefore, the CuNPs can form necking between each other [14] to create effective electron transfer network. However, the connection between CuNPs is still limited because the surrounding PVP did not decompose. Appendix A shows that the PVP started to decompose at ~330 °C. When the energy intensity increased to 3.23 J/cm^2^, the temperature of the copper film is estimated to be ~330 °C, close to the decomposition temperature of PVP.

However, thermal cracks occur when too much energy density is given. As shown in Figure 3, as the IPL energy intensity increases to 3.23 J/cm^2^, the resistivity rises again due to cleaves or cracks, and one can observe obvious chips and holes in the sintered thin films. Because the radiation energy is absorbed from top of copper film, the temperature on the top surface of the sintered thin films are higher than that of the substrate. Because the substrate temperature only increase slightly (~20 °C) [28,29,30,31] after IPL treatment, there exists a temperature difference ~300 °C between the sintered copper film and the substrate. As the result, the large thermal expansion difference between the copper film and the substrate leads to serious crack formation.

### 3.4. Multi-Pulse IPL Sintering

To avoid crack formation in the IPL sintering process, a multiple-pulse sintering process is adopted to reduce thermal yield stress across the sintered copper film. A series of pulses are applied with a fixed off time of 2 s between individual pulses (Figure 4a). The resistivity decreased with more IPL pulses (Figure 4b) and can reach a low value of 10^−5^ Ω·cm after five pulses. This decrease in resistivity indicates that the given energy can be properly accumulated in the sintered films with more uniform temperature distribution across the film [20]. However, cracks occur (Figure 5a) again when the total given energy is larger than 11 J/cm^2^, and leads to an increase in resistivity. To further investigate the sintering characteristics, the copper films treated with different number of pulses were observed by SEM (Figure 5b). The CuNPs obviously melted with more than four IPL pulses and show a compact or fused microstructure. Thus, these CuNPs need sufficient energy to melt or to form connections between particles. This multiple-pulse procedure can effectively sinter CuNP thin films without serious cracking, however, this process also leads to immense energy waste, and thus a more efficient methodology is needed.

### 3.5. Addition of Cupric Oxide (CuO) Nanoparticles

In order to increase photonic energy absorption, a photonic additive, cupuric oxide nanoparticles, is added into the CuNP ink. Copper oxide nanoparticles have been shown to absorb IPL sufficiently [15], and can be reduced to pure copper after sintered with the presence of reductant, such as PVP. To enhance the energy absorbance and to build connections between CuNPs, copper oxide nanoparticles (CuONP) of 50 nm diameter is added in the ink. As shown in Appendix A, CuONP has a broad absorption peak at ~670 nm, which is close to that of the IPL light source. Thus, the ink containing CuONP is expected to absorb energy more efficiently. Moreover, the smaller-size CuONP can fill in the voids between CuNPs. Samples with different CuONP/CuNP weight ratios are sintered by a single IPL pulse with an energy density of 2.77 J/cm^2^. As shown in Figure 6a, the optimal CuONP/CuNP ratio is 1/80, which yields in a lowest resistivity of 40 μΩ·cm, which is lower than that of CuNP ink after four pulses. The lower resistivity with CuONP indicates that the CuONPs are recovered to copper and fill in the void to build better connection between CuNPs. As shown in the inset SEM images, when the weight ratio increases to 1/80, CuONPs are fused significantly between large copper nanoparticles and build great connections. With the optimal weight ratio at 1/80, the samples can be sintered with a single pulse at an intensity of 3.1 J/cm^2^ to reach a lowest resistivity of 6.5 μΩ·cm, or 25% conductivity of bulk cooper. However, cracks (Appendix A) occur again when the sintering intensity is more than 3.23 J/cm^2^, and result in lower conductivity. At the optimal sintering conditions, the copper films exhibit a compact microstructure. SEM images (Appendix A) show that nanoparticles fused with each other closely after copper oxide addition. The cross-sectional SEM images (Figure 7a) also indicate that the copper films after addition of CuO have more compact microstructure with less voids. A larger peak height ratio between (111) and (200) after the addition of CuO particles is observed in the XRD patterns (Figure 7c), indicating a better crystalline orientation. [32] Besides the better conductivity performance, this method also leads to better energy utilization. As shown in Table 1, the sintering energy density used here (3.08 J/cm^2^) is much lower than those in the literature, but with a comparable conductivity.

### 3.6. Conductive Patterns Preparation

The CuONP/CuNP ink can be inkjet printed to form conductive tracks for electrical devices. As shown in Figure 8, the printed tracks on PET sheet becomes fairly conductive after IPL sintering, and can be used by interconnect conductors with light emiting doide (LED) lighting. The tracks are flexible and remain conductive under bending conditions. The resistance of the printed tracks are fairly stable, less than 20% difference, after 10,000 bending cycles. This low resistance variation indicates great adhesion on PET and good conductivity of the printed track. Thus, this ink can be further extended to other electronic applications.

## 4. Conclusions

In this study, an ink formulation is developed to prepare conductive copper thin films with compact structure by using intense pulsed light (IPL) sintering. First, a cuprous oxide shell was synthesized over copper nanoparticles (CuNP) of 200 nm. This cuprous oxide shell can be reduced by IPL and fused to form connection between the copper nanoparticles. However, thermal yield stress after strong IPL sintering results in cracks of conductive copper film. Thus, a multiple pulse sintering with an off time of two seconds was needed to reach a low resistivity of 10^−5^ Ω·cm. To increase the light absorption efficiency and to further decrease voids between CuNPs in the copper film, cupric oxide nanoparticles (CuONP) of 50 nm were added. These CuONPs were reduced to copper with a single pulse IPL and fused with the surrounding CuNPs. With an optimal CuNP/CuONP weight ratio of 1/80, the copper film resistivity can be as low as 6×10^−6^ Ω·cm, ~25% conductivity of bulk copper, with a single sintering energy at 3.08 J/cm^2^. This ink can be printed on flexible polyimide films as conductive tracks with great adhesion and shows great potential for flexible electronic applications.

## Figures and Tables

**Figure 1 nanomaterials-09-01071-f001:**
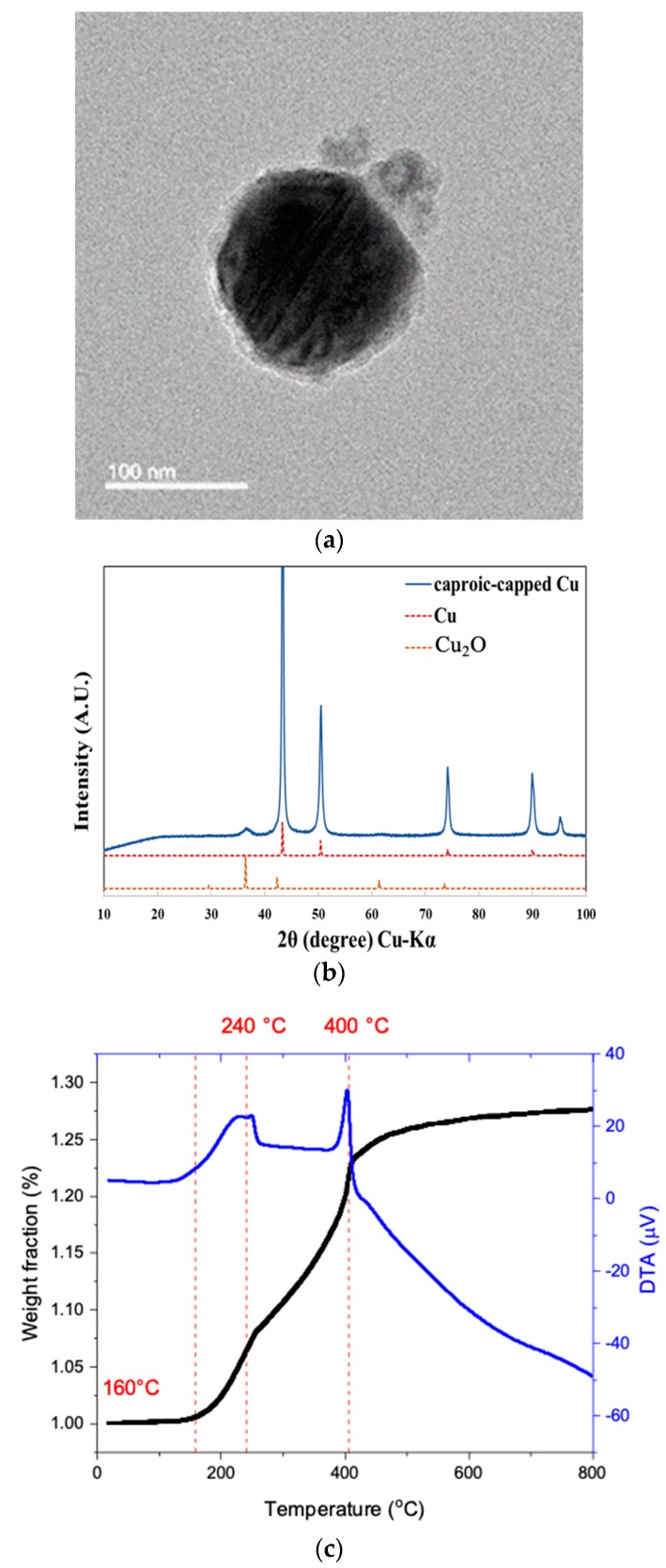
(**a**) Transmission electron microscope (TEM), (**b**) X-ray diffraction (XRD) and (**c**) TG-DTA curve of the synthesized copper nanoparticles.

**Figure 2 nanomaterials-09-01071-f002:**
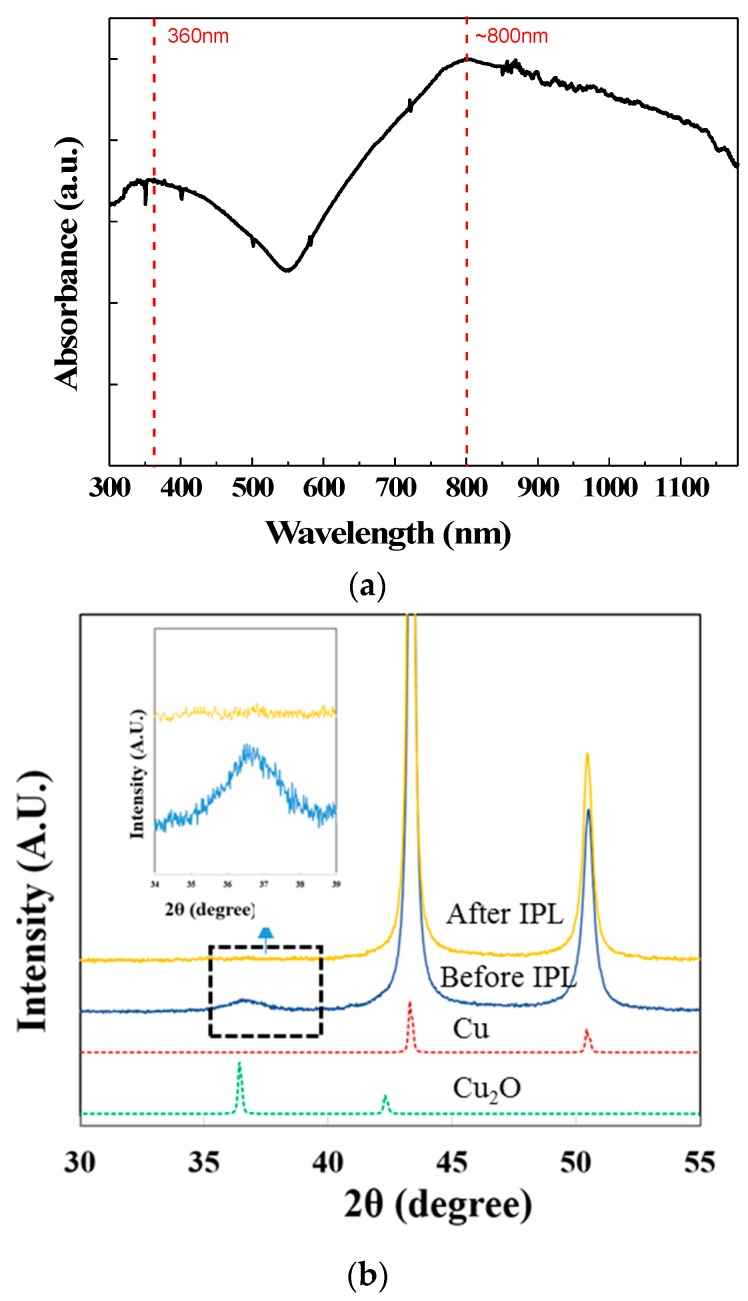
(**a**) UV-vis spectrum of the synthesized copper nanoparticles (CuNPs) and (**b**) the XRD patterns before and after intense pulsed light (IPL) sintering. An IPL energy density of 3.23 J/cm^2^ was used in the sintering.

**Figure 3 nanomaterials-09-01071-f003:**
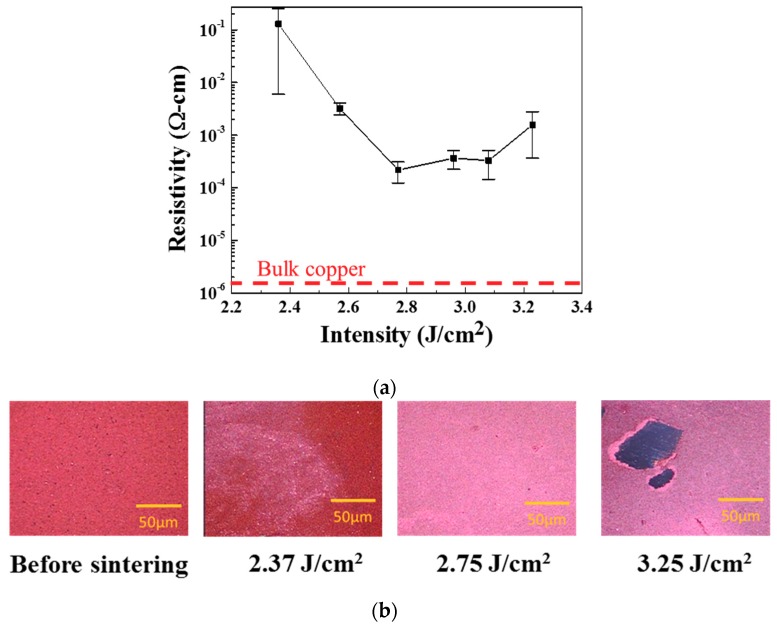
(**a**) Variation in resistivity of sintered copper films with single pulse energy intensity. (**b**) Optical images of copper film before and after sintering with various IPL energy intensities. The picture at 3.25 J/cm^2^ shows an example of a crack.

**Figure 4 nanomaterials-09-01071-f004:**
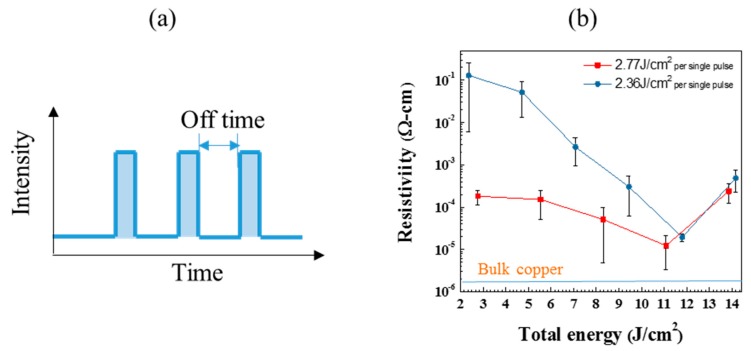
(**a**) Schematic diagram for multiple pulse IPL. The total area of pulses is defined as the total energy. (**b**) Resistivity of copper film sintered by multiple pulses. An off time of 2 s is used between IPL pulses.

**Figure 5 nanomaterials-09-01071-f005:**
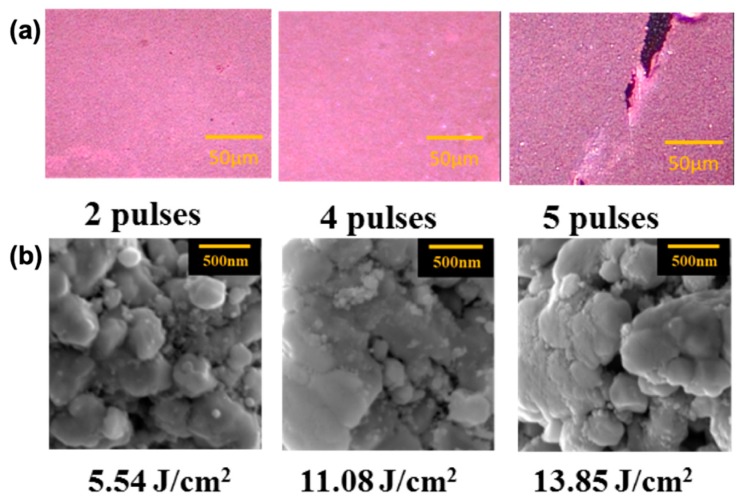
Optical (**a**) and scanning electron microscope (SEM) (**b**) images of sintered copper films by multiple pulses. Each single pulse has an energy density of 2.77 J/cm^2^.

**Figure 6 nanomaterials-09-01071-f006:**
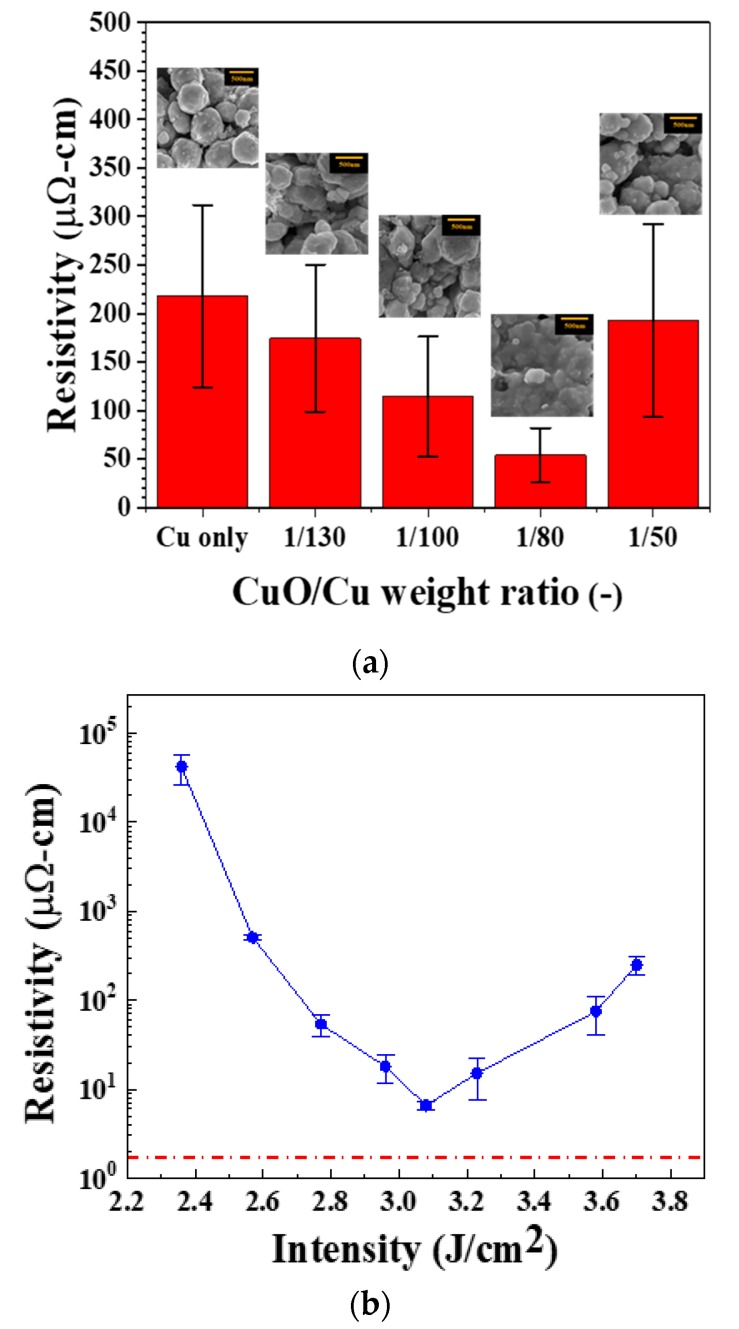
(**a**) Resistivity vs weight ratio of copper film, cupric oxide nanoparticles (CuONP)/CuNP. A single pulse with an energy density of 2.77 J/cm^2^ is used. (**b**) Resistivity of CuONP/CuNP (1/80 weight ratio) film sintered by single pulse IPL at various energy density.

**Figure 7 nanomaterials-09-01071-f007:**
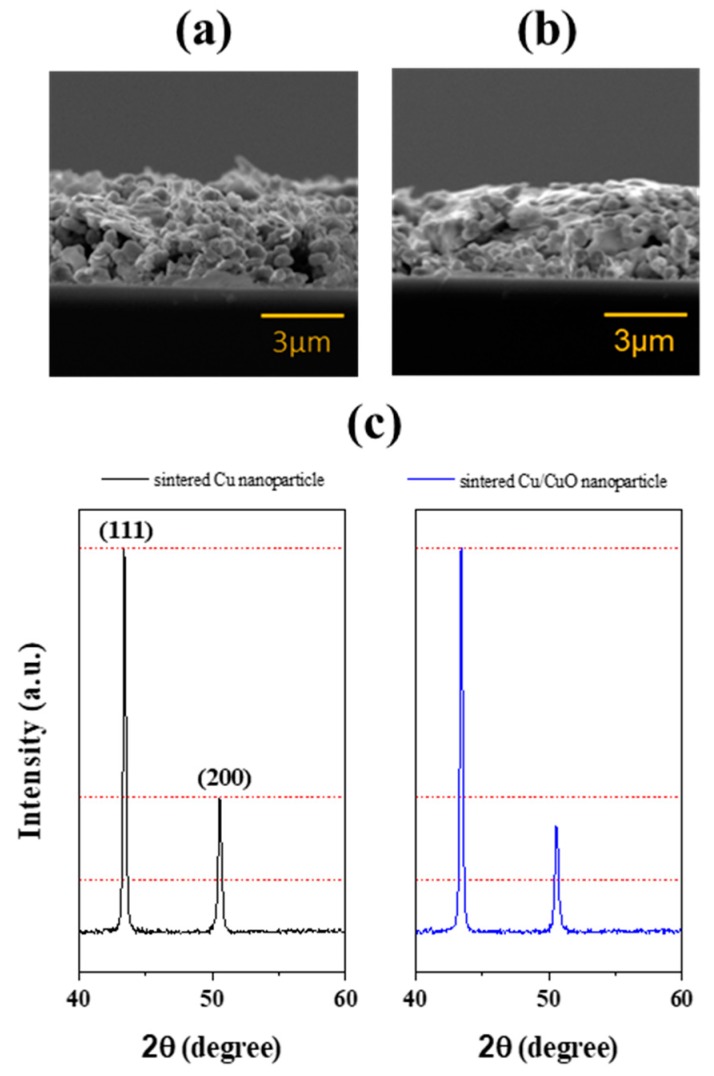
Cross section SEM images of (**a**) sintered Cu film and (**b**) sintered Cu/CuO film, and (**c**) XRD patterns comparison.

**Figure 8 nanomaterials-09-01071-f008:**
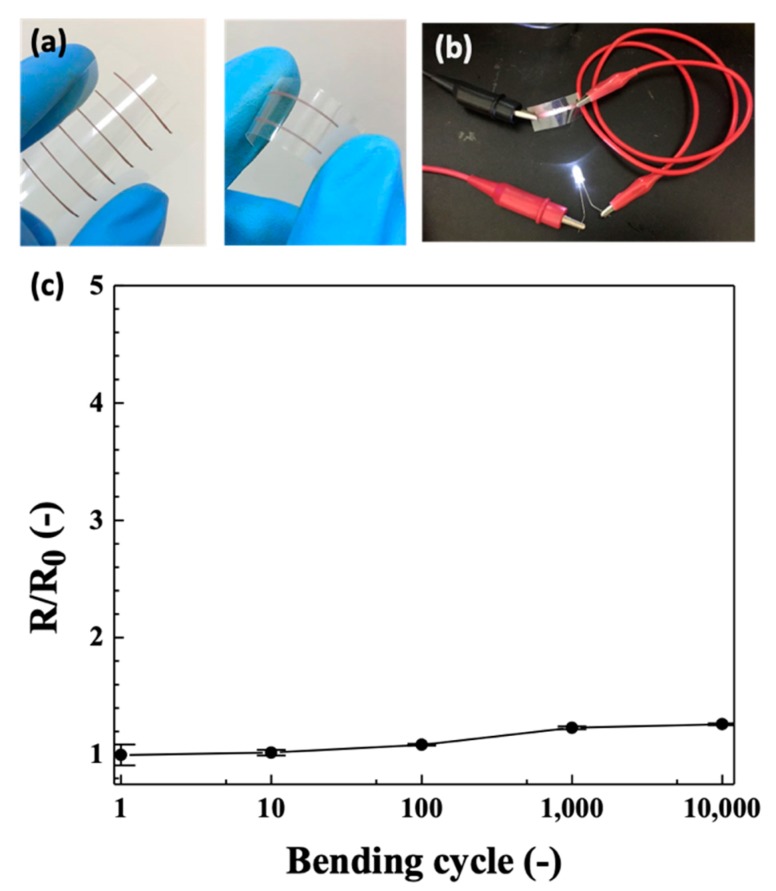
(**a**) Images of printed copper tracks on PET by using CuNP/CuONP ink. (**b**) LED light connected with the printed tracks. (**c**) Bending test for the printed tracks. A bending radius of 1 cm is used in this test.

**Table 1 nanomaterials-09-01071-t001:** Comparison of copper thin film from IPL sintering.

Reference	Particle Composition	Sintering Methods	Substrate	Resistivity (μΩ·cm)
Ref. [10]	CuO	7.48 J/cm^2^	Poly(ethylene terephthalate) films (PET)	5.5
Ref. [13]	CuNPs (20–50 nm)	12.5 J/cm^2^ + deep UV (30 mW)	PI (Kapton polymide)	7.6
Ref. [20]	Cu nanoparticles (<100 nm)	8 J/cm^2^, 1 ms duration 30 pulses + 4 J/cm^2^	PI	6.9
Ref. [33]	Cu nanoparticle (20 ~ 50 nm) + microparticle (2 µm)	12.5 J/cm^2^	PI	72.8
Ref. [34]	Cu(NO3)_2_ + CuNPs	35 pulses, 30 ms duration. 7 J/cm^2^ + 1 pulse, 9 J/cm^2^	PI	7.6
Ref. [35]	Graphene + catalytic copper particle	@1100 °C under Ar/H_2_ environment	Glass	149.6
This work	Cu/Cu_2_O core-shell NPs (100 nm/20 nm) + CuO NPs (<50 nm)	3.08 J/cm^2^	PET	6.5

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
