# Peer review of "Sintering Copper Nanoparticles with Photonic Additive for Printed Conductive Patterns by Intense Pulsed Light"

_nanomaterials, 2019, doi:10.3390/nano9081071_

Reviewer 1 Report

Authors have presented work on inkjet printing of CuO NP.

CuO is a gas sensitive p-type semiconductor and as such the resistivity will depend on the gas matrix. Did the authors check the influence of e.g. humidity?

Authors should add work being done in this regard to their state of the art discussion. After all inkjet printing of CuONP is not new. 

Author Response

The authors really thank the reviewer for the positive comments and suggestions.  The reviewer pointed out several important issues in the manuscript, and our answers are listed as follows.

 1.       CuO is a gas sensitive p-type semiconductor and as such the resistivity will depend on the gas matrix. Did the authors check the influence of e.g. humidity?

 Thank you very much for pointing out this important issue! We didn't check the influence of gas sensor. From the XRD pattern, CuO nanoparticles were converted to Cu after the sintering process, and peaks of CuO were not observed. Thus, we did not expect the copper conductive film to show properties of p-type semiconductor. 

 2.      Authors should add work being done in this regard to their state of the art discussion. After all inkjet printing of CuONP is not new. 

 Thank you very much for the suggestions! CuO nanoparticles were used for inkjet printing with IPL for a long time, but high energy intensity is generally needed. To address this issue regarding the CuO nanoparticles, we added a few citations for inkjet CuO inks and described the inkjet printing of CuONP on page 2 as follows.

 “…especially coupled with UV or infrared lights, [19]and IPL-sintered tracks can exhibit great conductivity close bulk copper. Moreover, it has been found that inkjet printed copper oxide nanoparticles can absorb IPL to yield locally high temperatures and react with the reduction agent into conductive copper thin films.[12-14]On the other hand, the intense local heating during the IPL process can cause film damages due to large thermal yield stresses. ….”

 12.   Öhlund, T.; Schuppert, A.K.; Hummelgard, M.; Bäckström, J.; Nilsson, H.-E.; Olin, H.k. Inkjet fabrication of copper patterns for flexible electronics: Using paper with active precoatings. ACS applied materials & interfaces 2015, 7, 18273-18282.

13.   Kang, H.; Sowade, E.; Baumann, R.R. In Photonic sintering of inkjet printed copper oxide layer, NIP & Digital Fabrication Conference, 2013; Society for Imaging Science and Technology: pp 494-497.

14.   Krcmar, P.; Kuritka, I.; Maslik, J.; Urbanek, P.; Bazant, P.; Machovsky, M.; Suly, P.; Merka, P. Fully inkjet-printed cuo sensor on flexible polymer substrate for alcohol vapours and humidity sensing at room temperature. Sensors 2019, 19, 3068.

Reviewer 2 Report

Chung and co-workers has presented a serious work demonstrating high performance in conductive inks based on copper nano-particles. The results are compared with previous achievements in literature a provide valuable technical information for researchers in this topic. Moreover, the author provide detailed structural and morphological characterization of the samples and details enough for the reader to reproduce their work. For all this reasons, I should recommend the manuscript for publication in “nanomaterials”, with just minor changes:

1)     The introduction is clear and focus the reader on the topic. However, there is a wide range of conductive inks based on different approaches and materials.

 The results of the manuscript would be also of interest for researchers of other fields. For examples, development of e-beam and photo-resists. E.g: LANGMUIR  26   2825-2830   (2010); or MICROELECTRONIC ENGINEERING  87   1147-1149  (2010).

 Quantitative comparison of among IPL sintering in Table 1 really points out the performance of the method. A qualitative discussion between IPL sintering and other methods and applications would be quite interesting, for example [2D MATERIALS  4  025088 (2017).

 And in general, reviewing bibliography and expanding the scope of the references would make the manuscript more interested for a non-specialized reader.

 2)     More details about the set-up for the 4-probe measurements would be desirable, as conductivity soft-materials could vary in the contracts used for electrical characterization. Probably, your samples are as conductive as in Ref 10 just they are carrying out measurements in better conditions.

 3)     Please, improve the quality of the figures. For example, in Fig. 1 without label (c) looks like elongated, like a print screen, with the axis titles pixelated.

4)     Figure 8 (c). Put Y axis in the proper scale.

 5)     I guess that Fig. S2 correspond to thin film measurements, doesn’t it? Please, let it clear.

 I hope these comments would help to improve your manuscript.

Author Response

The authors really thank the reviewer for the positive comments and suggestions.  The reviewer pointed out one important issue in the manuscript, and our answer is listed as follows.

 1.       The introduction is clear and focus the reader on the topic. However, there is a wide range of conductive inks based on different approaches and materials. 

The results of the manuscript would be also of interest for researchers of other fields. For examples, development of e-beam and photo-resists. E.g: LANGMUIR 26   2825-2830   (2010); or MICROELECTRONIC ENGINEERING  87  1147-1149  (2010). 

Quantitative comparison of among IPL sintering in Table 1 really points out the performance of the method. A qualitative discussion between IPL sintering and other methods and applications would be quite interesting, for example [2D MATERIALS  4 025088 (2017).  And in general, reviewing bibliography and expanding the scope of the references would make the manuscript more interested for a non-specialized reader.

        Thank you very much for the great suggestions! We cited the suggested reference in the introduction on page 2 as follows.

        “…To fabricate conductive thin film patterns, there are several ink-based approaches, such as inkjet printing [9], transfer printing and e-beam lithography [10,11]. …”

 9.     Shen, W.; Zhang, X.; Huang, Q.; Xu, Q.; Song, W. Preparation of solid silver nanoparticles for inkjet printed flexible electronics with high conductivity. Nanoscale 2014, 6, 1622-1628.

10.   Marqués-Hueso, J.; Abargues, R.; Canet-Ferrer, J.; Agouram, S.d.; Valdés, J.L.s.; Martínez-Pastor, J.P. Au-pva nanocomposite negative resist for one-step three-dimensional e-beam lithography. Langmuir 2009, 26, 2825-2830.

11.   Marques-Hueso, J.; Abargues, R.; Canet-Ferrer, J.; Valdes, J.; Martinez-Pastor, J. Resist-based silver nanocomposites synthesized by lithographic methods. Microelectronic Engineering 2010, 87, 1147-1149.

 We also added the citations and described the IPL and others sintering methods on table 1 as follows.

Reference

Particle Composition

Sintering methods

Substrate

Resistivity

(μΩ-cm)

Kang et al. 2014

Ref. [10] 

CuO

 7.48 J/cm2

PET

5.5

Hwang et al., 2016

Ref [13]

CuNPs (20-50 nm) 

12.5 J/cm2+deep UV (30mW)

PI

7.6

Hwuag et al. 2015

Ref. [20]

Cu nanoparticles(<100nm)< strong="">

8J/cm, 1ms duration 30 pulses+ 4J/cm2

PI

6.9

Joo et al. 2014

Ref. [30]

Cu nanoparticle(20~50nm)

+ microparticle (2µm)

12.5 J/cm2

PI

72.8

Jeon et al., 2016 

Ref. [31]

Cu(NO3)2 + CuNPs 

35 pulses, 30 ms duration. 7 J/cm+ 1 pulse, 9 J/cm2

PI

7.6

Marchena et al., 2017

Ref.[32]

Graphene

+ catalytic copper particle

@1100 °C under Ar/H2environment

glass

149.6

This work

Cu/Cu2O core-shell NPs(100nm/20nm)+CuO NPs(<50nm)< strong="">

3.08 J/cm2

PET

6.5

 32.   Marchena, M.; Song, Z.; Senaratne, W.; Li, C.; Liu, X.; Baker, D.; Ferrer, J.C.; Mazumder, P.; Soni, K.; Lee, R. Direct growth of 2d and 3d graphene nano-structures over large glass substrates by tuning a sacrificial cu-template layer. 2D Materials 2017, 4, 025088.

 2.       More details about the set-up for the 4-probe measurements would be desirable, as conductivity soft-materials could vary in the contracts used for electrical characterization. Probably, your samples are as conductive as in Ref 10 just they are carrying out measurements in better conditions.

 Thank you very much for the great suggestions! We described about the measurements of conductive thin films as follows.

 “…The sheet resistance (R) of copper films was measured by a four-point probe (Keithlink, TG-2). Copper thin film samples (~25px2) were placed on a flat surface and gently touched with the probe. At least 5 sample points were sampled to collect the sheet resistance data. …”

3.       Please, improve the quality of the figures. For example, in Fig. 1 without label (c) looks like elongated, like a print screen, with the axis titles pixelated.

 Thank you very much for the great suggestions! We revised some figures 1(c) so that the figures and images in the manuscript can be more clearly displayed. 

4.       Figure 8 (c). Put Y axis in the proper scale.

 Thank you very much for the great suggestions! We revised figures 8(c) so that the Y axis in the figure can more clearly shown.

 5.       I guess that Fig. S2 correspond to thin film measurements, doesn’t it? Please, let it clear.

 Thank you very much for the great suggestions! We used nanomaterial solution for the UV vis measurement. The description was added in the figure caption to make it clear to readers. 

 Figure S2. UV-vis absorbance spectrum comparison of (a) CuNP, (b)CuONPs and (c) CuONP/CuNP mixture with a weight ratio of 1/80. All the spectra were obtained by using the prepared nanomaterial ink samples in solution form.

Reviewer 3 Report

The manuscript is well organized and well written and so can be accepted for publication after a few minor changes. In my opinion figure 1c should be improved and could present the onset point.

Author Response

Reviewer #3

 The authors really thank the reviewer for the positive comments and suggestions.  The reviewer pointed out one important issue in the manuscript, and our answer is listed as follows.

 1.       The manuscript is well organized and well written and so can be accepted for publication after a few minor changes. In my opinion figure 1c should be improved and could present the onset point.

 Thank you very much for the great suggestions! We revised figure 1(c) so that the onset temperatures can be clearly displayed.

Reviewer 4 Report

The paper deals with conductive copper films made of Cu nanoparticles and with their sintering performed by intense pulse light (IPL) technology. The authors develop a simple ink formulation to improve light absorption efficiency in IPL.  

The paper is concise and quite clear. The study is interesting for the application in flexible electronics.

The paper could be suitable for the publication after a major revision. There are some points to address and some info to add.

“The core-shell CuNPs can be quickly sintered after IPL irradiation. The UV-vis spectrum of CuNPs (Figure 2(a)) shows two main absorption peaks at 360 nm and 800 nm, which correspond to the Cu core particles and surrounding Cu2O particles.[24]” The peak at 800 nm is not clear. Why are the peaks so broad?

In figure 3 the resistivity is measured with large errors. Why such high incertitude in the measurements?

The caption of figure 3: Remove “I think  that larger images are better.” The authors could add that the picture at 3.25 J/cm^2 shows an example of a crack.

The authors mention “Polyimide films (PI, InTech Materials Co., Taiwan), Poly (ethylene terephthalate) films (PET, Universal film, Japan) and glass slides” which were cleaned by sonication in ethanol for 1 hour. However, it is not clear which substrate has been used for the samples they have presented. Also, they could better discuss the role of each of the considered substrates in the IPL process.

The thickness of Cu films is never given. Which is the control they have of the printed patterns? Is the formation of cracks related to the thickness of the samples? I think that the authors should also correlate the IPL to the film thickness.

Author Response

The authors really thank the reviewer for the positive comments and suggestions.  The reviewer pointed out one important issue in the manuscript, and our answer is listed as follows.

 1.       “The core-shell CuNPs can be quickly sintered after IPL irradiation. The UV-vis spectrum of CuNPs (Figure 2(a)) shows two main absorption peaks at 360 nm and 800 nm, which correspond to the Cu core particles and surrounding Cu2O particles.[24]” The peak at 800 nm is not clear. Why are the peaks so broad?

 Thank you very much for the great suggestions!The peak at 800 nm shows the structure of Cu2O particles. We are sorry that in the previous manuscript, the range beyond 800 nm was cut. We now put the spectrum in the full range of 300-1200 nm to show the two peaks. Both of peaks are broad, because the size of copper nanoparticles has a wide distribution. We added a sentence and a reference to address this issue.

 “…surrounding Cu2O particles.[30] These two peaks are broad due to the wide size distribution of CuNPs.Thus, these particle…”

2.       In figure 3 the resistivity is measured with large errors. Why such high incertitude in the measurements?

 Thank you very much for the great suggestions! The large incertitude is caused by the non-uniformity of the sintered thin films. At high IPL energy, because the probe might touch cracks on the thin films in 4-probe measurements, and the resistance value changes greatly. At low IPL energy, the films were not totally recovered to copper, and thus the resistance around the edges of the samples, which has a lower sintering temperature, were not very good. Because we needed to sample several spots on the samples, therefore there exist a large resistance differences at these sampling points.  However, this problem does not occur with proper IPL energy, which leads to uniform thin films, so the value of the stabilizing resistance can be measured. To address this issue, we added a sentence in the discussion regarding figure 3.

 “…with single pulse.At low IPL energy of 2.37 J/cm2, the CuNPs are not totally sintered and thus a large measurement incertitude is observed (Figure 3 (a)).  As IPL energy density increases, CuNPs are well sintered and with uniform conductivity and shows the lowest resistivity of …”

 3.       The caption of figure 3: Remove “I think that larger images are better.” The authors could add that the picture at 3.25 J/cm^2 shows an example of a crack.

 Thank you very much for the great suggestions! We revised sentence in figure 3 as follows.

 “…before and after sintering with various IPL energy intensities.The picture at 3.25 J/cm2shows an example of a crack.…”

 4.       The authors mention “Polyimide films (PI, InTech Materials Co., Taiwan), Poly (ethylene terephthalate) films (PET, Universal film, Japan) and glass slides” which were cleaned by sonication in ethanol for 1 hour. However, it is not clear which substrate has been used for the samples they have presented. Also, they could better discuss the role of each of the considered substrates in the IPL process.

 We apologize for the wrong descriptions about the substrate used in our experiments and truly thank the reviewer for the suggestion! In this study, we used only PET as the substrate for the conductive copper films. Thus, we removed all the descriptions about glass slides and PI films in the experiment section. 

 5.       The thickness of Cu films is never given. Which is the control they have of the printed patterns? Is the formation of cracks related to the thickness of the samples? I think that the authors should also correlate the IPL to the film thickness.

 Thank you very much for the suggestions! In this study, the copper films were controlled by using the same printing parameters to have the same thickness of 3 mm as shown in figure 7(a-b). The formation of cracks is mostly due to thermal yield stress from high IPL energy as shown in figure 3(b). To address this issue, some sentences were added in the experimental sections:

 “… All the samples were printed with the same printing parameters to have the same deposition rate and thus the same thickness was expected.The printed …”

Round  2

Reviewer 4 Report

The authors have made several changes in the paper. They have addressed all the points raised by the reviewer. The paper can be accepted for the publication.